# Adherence to Pro-Vegetarian Food Patterns and Risk of Oesophagus, Stomach, and Pancreas Cancers: A Multi Case–Control Study (The PANESOES Study)

**DOI:** 10.3390/nu14245288

**Published:** 2022-12-12

**Authors:** Alejandro Oncina-Cánovas, Sandra González-Palacios, Leyre Notario-Barandiaran, Laura Torres-Collado, Antonio Signes-Pastor, Enrique de-Madaria, Miguel Santibañez, Manuela García-de la Hera, Jesús Vioque

**Affiliations:** 1Instituto de Investigación Sanitaria y Biomédica de Alicante, Universidad Miguel Hernández (ISABIAL-UMH), 03010 Alicante, Spain; 2Unidad de Epidemiología de la Nutrición, Departamento de Salud Pública, Historia de la Ciencia y Ginecología, Universidad Miguel Hernández (UMH), 03550 Alicante, Spain; 3CIBER Epidemiología y Salud Pública (CIBERESP), Instituto de Salud Carlos III, 28034 Madrid, Spain; 4Servicio de Gastroenterología, Hospital General Universitario Dr. Balmis, 03010 Alicante, Spain; 5Grupo de Investigación de Salud Global, Instituto de Investigación Marqués de Valdecilla (IDIVAL), Universidad de Cantabria, 39011 Santander, Spain

**Keywords:** pro-vegetarian food patterns, oesophagus, stomach, pancreas, cancer, food quality

## Abstract

We aimed to evaluate the association between three previously defined pro-vegetarian (PVG) food patterns and the cancers of the oesophagus, stomach, and pancreas in a multi case–control study. We analyzed data from a multi-case hospital-based study carried out in two Mediterranean provinces in Spain. A total of 1233 participants were included in the analyses: 778 incident cancer cases, histologically confirmed (199 oesophagus, 414 stomach, and 165 pancreas) and 455 controls. A dietary assessment was performed using a validated food frequency questionnaire (FFQ). Three PVG food patterns (general, healthful, and unhealthful) were estimated using 12 food groups for the general PVG (gPVG), scoring positive plant-based foods and negative animal-based foods, and 18 food groups, for the healthful (hPVG) and unhealthful (uPVG) food patterns. Multinomial logistic regression was used to estimate relative risk ratios (RRR) and confidence intervals (95% CI) for quintiles of adherence to PVG patterns and as a continuous variable. The RRR (95% CI) for the highest vs. the lowest quintile of gPVG were, RRR = 0.37 (0.32, 0.42) for the oesophagus, RRR = 0.34 (0.27, 0.43) for the stomach, and RRR = 0.43 (0.35, 0.52) for pancreas cancer. For the hPVG, the RRR were RRR = 0.72 (0.58, 0.90) for the oesophagus, RRR = 0.42 (0.34, 0.52) for the stomach, and RRR = 0.74 (0.59, 0.92) for pancreas cancer. The uPVG was associated with a higher risk of stomach cancer RRR = 1.76 (1.42, 2.18). Higher adherence to gPVG and hPVG food patterns is associated with a lower risk of oesophageal, stomach, and pancreas cancers, while a higher adherence to a uPVG food pattern is associated with a higher risk of stomach cancer.

## 1. Introduction

Cancer remains one of the world’s leading causes of death. The International Agency for Research on Cancer (IARC) estimates that in 2020 there were approximately 20 million new cases and 9.9 million cancer deaths worldwide. After lung and breast cancer, digestive cancers are the major cause of morbidity and mortality by cancer, especially in men [1]. The cancer mortality statistics in 2020 in Spain evidenced that colorectal cancer was the second cause of death after lung cancer, pancreas cancer the third, and stomach and oesophagus cancers in lower positions, although still causing a large number of deaths despite their decreasing trends [2]. Oesophageal, stomach, and pancreas cancers have poor survival and their aetiology is insufficiently known.

Alcohol and tobacco consumption have been associated with an increased risk of these cancers although the results are not fully consistent [3,4,5]. The role of diet has also been investigated. Previous studies have explored the association between the intake of specific nutrients or foods and the risk of these cancers. Vitamin C may have a protective effect on oesophageal, gastric, and pancreatic cancers although the results are not fully consistent [6,7]. The consumption of high-salt and salt-preserved foods have also been associated with an increased risk of stomach cancer [8,9]. Hot beverages such as tea and, especially, mate tea, have been classified as “probably carcinogenic to humans” (Group 2A) increasing the risk of oesophageal cancer when they exceed 65 °C [10]. Consumption of red and processed meats is considered a risk factor for colorectal cancer and has also been associated with an increased risk of stomach cancer [11,12,13]. Other exposures, like mycotoxin contamination, have also been studied in these cancers. In this sense, some foods such as cereals or nuts could be potential dietary sources of aflatoxins, a type of mycotoxins, that have been associated with oesophageal cancer [14].

One difficulty in studying the role of specific nutrients and foods is that people do not consume foods in isolation, but rather a combination of them. Thus, the study of food patterns has been proposed as a more complex and realistic alternative for assessing the combined effect of diet. One of the most studied food patterns has been the Mediterranean diet (MD). A recently published systematic review [15] has shown that greater adherence to an MD is associated with a lower risk of gastrointestinal cancers, although the overall certainty of the evidence was considered ‘low’ or ‘very low’. Another food pattern that has gained popularity in the last decade are plant-based diets (PBDs). In this sense, vegetarian diets have been recognized as one of the most important PBDs [16]. Since vegetarian diets are restrictive patterns, in which animal food is excluded to a greater or lesser extent, other more flexible dietary options have been proposed. Hence, a general pro-vegetarian (gPVG) food pattern emerged a few years ago as a novel option in which, instead of excluding foods of animal origin, plant-based foods were prioritized, scoring them positively, whereas animal foods were scored negatively, as described in more detail later [17]. Then, this pattern was adapted by Satija et al. including what we know from the scientific literature about plant-based foods’ healthiness, formulating two derivations, one healthful (hPVG) and another unhealthful (uPVG) [18]. There is limited evidence on the relationship between these three PVG food patterns and cancer. Recently Jihye Kim et al. examined the association between three plant-based dietary indices (PDI), similar to the PVG food patterns described above, and observed in a sample of 118,577 South Korean adults, that greater adherence to unhealthful PDI (uPDI) was associated with an increased cancer risk of 1.23 points [19]. In another study from the US National Health and Nutrition Examination Survey (NHANES), the authors observed a 32% lower overall cancer risk among the most adherent participants to an overall PDI [20]. Although there exists some evidence for an association of these patterns to specific cancers, such as breast [21], prostate [22], or basal cell carcinoma [23], there remains scarce knowledge about the association with gastrointestinal cancers.

Thus, we aimed to investigate the association between three a priori defined PVG food patterns (gPVG, hPVG, and uPVG) and cancers of the oesophagus, stomach, and pancreas, in the PANESOES multi case–control study carried out in Spain.

## 2. Materials and Methods

### 2.1. Study Design and Population

The PANESOES study is a hospital-based, multi case–control study aimed to explore the effect of major lifestyles, including diet, on the risk of three types of gastrointestinal cancer: oesophageal, stomach, and pancreatic. More details of the study have been described previously [24,25,26,27]. Briefly, the PANESOES study was aimed to recruit 200 cases of oesophageal cancer, 400 cases of stomach cancer, 200 cases of pancreatic cancer, and 450 controls frequency matched to cases by sex, age, and province (Alicante and Valencia). The participants selected were Spanish-speaking women and men aged 30–80 years, who were recruited after hospitalization between January 1995 and March 1999. The hospitals recruiting the study population were 9 in the provinces of Alicante (Hospital General, Hospital Clínico de San Juan, Hospital de Elche, and Hospital Comarcal de la Vega Baja) and Valencia (Hospital Clínico Universitario, Hospital La Fe, Hospital Dr. Peset, and Hospital Arnau de Vilanova). Research protocols developed for the study were approved by the local ethics and/or research committees of the participating hospitals and the university (AUT.DSP.JVL.04.21).

After excluding those participants with missing information for the variables of interest and/or without the necessary information to construct the PVG food patterns, 1233 participants (840 men and 393 women), corresponding to 199 cases for oesophageal cancer, 414 cases for stomach cancer, 165 cases for pancreatic cancer, and 455 controls, were included in the present analysis, for which there was cyto-histological confirmation and/or ample clinical evidence. A total of 455 controls were selected from the same hospitals and with the same characteristics as the cases matched by frequencies of sex, age (3 categories: 30–59; 60–69; and 70–80), and province (Alicante/Valencia). The diagnoses of the controls were a priori unrelated to the exposure of interest (hernias: 34%, degenerative osteoarthritis: 21%, fractures/accidents/orthopaedic processes: 19%; appendicitis: 6%, and other diagnoses: 20%).

All participants, prior to data collection, were informed of the aims of the study and agreed by informed consent to complete the interview.

### 2.2. Dietary Intake and Pro-Vegetarian Food Patterns

Information on usual dietary intake was collected by trained interviewers using a semi-quantitative food frequency questionnaire (FFQ), based on the Harvard questionnaire [28], which we developed and validated using four 1-week dietary records in an adult population in Valencia [29]. During the hospital interview, participants in the study were asked how often, on average, they had consumed each food item of the FFQ over a whole year, referred to 5 years prior to the hospital interview. The FFQ included 93 food items and 9 options for frequency of consumption of each food, ranging from ‘Never or less than 1 time per month’ to ‘6 or more times per day’.

The PVG food patterns were created following, in the case of the gPVG, the methodology proposed by Martínez-González [17], and in the case of hPVG and uPVG food patterns, the Satija method [18]. Table 1 shows the 18 food groups used to create the PVG food patterns and how they scored: 13 plant food-based (vegetables, fruits, legumes, whole grains, refined grains, potatoes, fries and chips, nuts, olive oil, tea and coffee, fruit juices, sugar-sweetened beverages, and sweets and desserts), and 5 animal food-based (meat and processed meats, animal fats for cooking, eggs, fish and seafood, and dairy products). Briefly, for the creation of the different PVG food patterns, the consumption in grams of each food group was adjusted for energy intake using the residual method [30]. Then, the energy-adjusted consumption in grams of each food group was divided into quintiles. Next, the scores of the quintiles of all food groups were added together according to each PVG food pattern. In the gPVG food pattern, seven plant-based food groups scored positively (vegetables, fruits, legumes, grains, including whole and refined, potatoes, including boiled and baked and fries and chips, nuts, and olive oil), with 5 being the highest score in the case of the higher consumption; and five animal origin food groups scored negatively (meat and processed meats, animal fats for cooking, eggs, fish and seafood, and dairy products) with the lowest consumption of these foods scoring 5 points. For the hPVG and uPVG versions, we added four new food groups (tea and coffee, fruit juices, sugar-sweetened beverages, and sweets and desserts) and we also considered the effect of refined and whole grains as well as the potatoes (boiled and roasted) and fries and chips. Finally, we obtained the total score for each participant by summing the points of the 12 food groups, for the gPVG food pattern, and 18 food groups, for the hPVG and uPVG versions. In this way, the score could remain between 12 points (minimum adherence) to 60 points (maximum adherence) in the case of the gPVG pattern, and between 18 points (minimum adherence) to 90 points (maximum adherence) in the case of hPVG and uPVG patterns.

### 2.3. Other Variables

The following information about different socio-demographic and lifestyle characteristics was also collected from participants: age (in years), sex (male or female), province (Alicante or Valencia), educational level (<primary, primary, >primary), tobacco consumption (never; former smoker; ≤24 cigarettes per day; >24 cigarettes per day), alcohol consumption (never, 1–24 g per day; 25–49 g per day; 50–99 g per day; >99 g per day), and energy intake (Kilocalories per day).

### 2.4. Statistical Analysis

We used multinomial logistic regression to estimate relative risk ratios (RRR) and 95% confidence intervals (95% CI) to explore the association between adherence to PVG food patterns by quintile of adherence and oesophagus, stomach, and pancreas cancer. We also explored this association per 1 additional point of adherence to each PVG food pattern.

Two models were adjusted, a first model included the matching variables age (years), sex (male, female), and province (Alicante, Valencia), and a second model additionally adjusted for variables previously described in the literature to be associated with these cancers, and also for variables that changed the association with exposure by 10% or more, once we excluded them from the model: educational level (<primary, primary, >primary), tobacco consumption (never; former smoker; ≤24 cigarettes per day; >24 cigarettes per day), alcohol consumption (never, 1–24 g per day; 25–49 g per day; 50–99 g per day; >99 g per day) and energy intake (Kilocalories per day).

Tests for the trend in the RRR across exposure strata were calculated for ordinal variables by using models that included categorical terms as continuous variables in a model with all the potential confounders. For trend tests, we used the likelihood ratio test statistic with one degree of freedom. Statistical significance was set at 0.05. All reported *p*-values are from two-sided tests. All analyses were performed with R version 4.0.3 (R Foundation for Statistical Computing, Vienna, Austria; http://www.R-project.org, accessed on 8 December 2022).

## 3. Results

Table 2 shows the distribution of cases and controls according to different socio-demographic and lifestyle characteristics. Educational level was comparable between cases and controls. Alcohol drinking and tobacco smoking were more prevalent in oesophageal cancer than in the other cases and controls. Oesophageal cancer cases showed the highest energy intake.

Table 3 shows multivariate analyses for the association between adherence to the gPVG food pattern (in quintiles of adherence and per 1 unit of additional adherence) and cancers of the oesophagus, stomach, and pancreas.

Compared with the lowest quintile of adherence to the gPVG, the highest adherence showed a 63% lower risk of oesophageal cancer, RRR = 0.37 (95% CI: 0.32, 0.42; *p*-trend = 0.01), a 66% lower risk of stomach cancer, RRR = 0.34 (95% CI: 0.27, 0.43; *p*-trend = 0.001), and a 57% lower risk of pancreas cancer, RRR = 0.43 (95% CI: 0.35, 0.52; *p*-trend = 0.01). When the association with adherence to the gPVG was evaluated as a continuous term, a 5 to 6% lower risk was observed for the three cancers for each additional unit of adherence.

Table 4 presents the association between the healthful PVG food pattern and the three cancers. The hPVG also showed a protective association for the three cancers with significant dose–response. Compared with the lowest quintile, the highest quintile of adherence was associated with a 28% lower risk of oesophageal cancer, a 58% lower risk of stomach cancer, and a 26% lower risk of pancreas cancer. Each additional unit of adherence to the hPVG was associated with a 5% lower risk of stomach cancer and a 2% lower risk of oesophagus and pancreas cancers.

In contrast to the inverse association observed for the other patterns, the unhealthful PVG was associated with a higher risk of stomach cancer (Table 5). The highest quintile of adherence to the uPVG showed a 76% higher risk of stomach cancer than the lowest quintile, RRR = 1.76 (95% CI: 1.42, 2.18; *p*-trend = 0.01). Each unit of additional adherence to the uPVG food pattern was associated with a 3% increased risk of stomach cancer risk, RRR = 1.03 (95% CI: 1.02, 1.05; *p*-trend = 0.01). No associations were observed between uPVG food pattern and oesophagus and pancreas cancer.

## 4. Discussion

This study suggests that a higher adherence to the general and healthful PVG food patterns is associated with a lower risk of oesophageal, stomach, and pancreatic cancers, while a higher adherence to the unhealthful PVG food pattern is associated with a higher risk of stomach cancer.

The PVG food patterns are a novel dietary model with potential future public health implications, although they are still understudied. To our knowledge, this is the first study that has evaluated the association between three PVG patterns, including hPVG and uPVG versions, and the risk of these three gastrointestinal cancers. Some prospective cohort studies in the US [20,31] and Korea [19] have evaluated the association between PBDs and overall cancer mortality, although the results were contradictory. A cohort study conducted in France with 42,544 adults reported a 34% lower risk of digestive cancers among those with higher adherence to a pro-plant dietary score with some similarities to our gPVG [32]. In a recent meta-analysis with information from 49 studies and with more than three million participants, it was reported that PBDs were, in general, protective against all types of digestive system cancers, although the definition of PBD was based on two very broad categories, diets excluding any meat, meat products, seafood, or food of animal origin (i.e., vegetarian and vegan diets, respectively) and diets characterized by a higher consumption of fruits, vegetables, legumes, and nuts rather than animal products [33]. Thus, our results may not be fully comparable to the results from this meta-analysis, and, therefore, more studies with a clearer definition and precise categorization of food patterns would be required to confirm the results.

There are several mechanisms that could explain the protective association observed for gPVG and hPVG food patterns and cancers of the oesophagus, stomach, and pancreas in our study. Plant-based foods included in these patterns are the main source of several key nutrients (eg., fiber, polyphenols). Dietary fiber is an important component of whole grains, fruits, and vegetables, and has been associated with a lower risk for several gastrointestinal cancers [34]. Fiber contains some phenolic compounds like ferulic acid that may have an antiproliferative effect on the cell cycle and this could help to prevent oesophagus cancer [35]. In the case of stomach cancer, dietary fiber may reduce the nitrite levels in the stomach [36]. These compounds can take the form N-nitroso and together with the addition of amines, would form nitrosamines, compounds classified as carcinogenic [37]. Finally, fiber may have some biological mechanisms for pancreas cancer prevention. Firstly, it may act by reducing carcinogen exposure in the intestinal lumen through the stool bulk effect. Secondly, it may modulate the microbiota through short-chain fatty acids production, and improve glucose homeostasis and insulin sensitivity, both related to tumor proliferation and anti-inflammatory effects. Thirdly, high fiber intake is associated with a healthy lifestyle and a behavioral factor that might reduce obesity risk, a well-known risk factor [38,39]. Plant-based foods are also a good source of polyphenols, whose anti-carcinogenic activity has been described for these three cancers in the literature [40]. Flavonoids, the most important family of polyphenols, are mainly found in fruits, vegetables, and legumes, and they may have effects on several cancer-related signaling pathways such as enhancing immunity, inhibiting oncogenic growth signaling pathways, and activating apoptosis [41]. Isoflavonoids, one of the most important sub-classes of flavonoids, might inhibit the growth of oesophageal squamous cell carcinoma cell lines [42]. The bioactivity described for flavonoids also could inhibit the growth of H. Pilory, a bacterium related to gastric cancer [43]. Ultimately, pancreas cancer may be prevented by the antidiabetic activity described for the flavonoids and lignans [44]. Other mechanisms may be related to the lower consumption of animal foods usually associated with these plant food patterns. For example, processed meats contain nitrites that, as we mentioned above, could form some carcinogenic substances like nitrosamines [37]. Other foods such as dairy products, especially milk, have also been reported to be associated with increased concentrations of insulin-like growth factor type I (IGF–1) [45]. This protein could induce tumor growth and metastasis through different signaling routes described in a previous study [46]. So, the synergistic effect of increasing some fresh vegetable consumption and reducing animal (especially processed) consumption could explain our findings for these cancers.

On the other side, the association between the uPVG food pattern and an increased risk of stomach cancer that we found in our study may be related to the combined effect of high-processed plant-based foods included in this pattern. Refined grains, potato chips, sugar-sweetened beverages, and sweets belong to the ultra-processed category and have been associated with an increase in overall cancer [47]. These food groups could act in several ways. On the one hand, they are dense in calories and have a lower satiety effect, and, consequently, their intake may lead to weight gain [48]. Obesity is a known risk factor for various types of cancer, including stomach cancer [49]. On the other hand, these processed foods are a good source of low-quality fats (saturated and trans-fatty acids), free sugars, and salt [47]. Trans-fatty acids have been linked to an increased risk of different types of cancer [50,51] and sugar is closely linked to obesity [52] (a major risk factor). Moreover, as we mentioned above, the habitual consumption of foods preserved with salt has been associated with gastric cancer [8,9]. Lastly, we are talking about food patterns, and it is important to note that these foods also act indirectly by displacing the consumption of other healthier options.

This study has some limitations. Firstly, the sample was limited, especially for oesophageal and pancreatic cancers, which may have reduced the statistical power to detect some associations. Nevertheless, dose–response and statistically significant associations were found for these cancers. The case–control study design is more susceptible to some biases, such as selection bias, although the participation rate was very similar in cases and controls (98%). The fact that diet was assessed five years before the interview might have caused misclassification bias, although the response was similar in cases and controls, and the 5-year reproducibility and validity of the FFQ we observed was satisfactory. In addition, although several confounding factors were considered in multivariable analyses, there may be other potential confounding factors that could influence the risk of developing the cancers of interest.

Our study also presents some strengths. Firstly, the association found between higher adherence to the three PVG food patterns and cancer risk after adjusting for several well-known exposure factors such as tobacco and alcohol consumption. Secondly, the use of the same protocol for the three cancers and the use of a well-structured questionnaire with a validated FFQ allowed us to construct the PVG food patterns that reinforced our findings. Finally, the strength of the associations and the existence of a dose–response effect also lent robustness to the results obtained.

## 5. Conclusions

In conclusion, this multi case–control study suggests that greater adherence to gPVG and hPVG food patterns rich in fruits and vegetables is associated with a lower risk of oesophageal, stomach, and pancreatic cancers. In contrast, a PVG food pattern that prioritizes less healthy foods like high-processed plant-based foods such as refined grains, potato chips, sugar-sweetened beverages, and sweets may be associated with a higher risk of stomach cancer and their consumption should be restricted. If confirmed by other studies, these findings could be a good alternative for making more precise public health recommendations based on healthy and unhealthy plant-based foods.

## Figures and Tables

**Table 1 nutrients-14-05288-t001:** Scoring criteria for each PVG food pattern.

Food Groups	Included Food Items	gPVG ^3^	hPVG	uPVG
Plant food groups ^2^				
1. Vegetables	Spinach, cabbage, cauliflower, broccoli, lettuce, endive, tomatoes, onion, carrot, pumpkin, green beans, eggplant, zucchini, cucumber, peppers, asparagus	Positive ^1^	Positive	Reverse
2. Fruits	Oranges, grapefruit, mandarin, banana, apple, pear, strawberries, cherries, peaches, apricots, fresh figs, watermelon, melon, grapes, canned fruit (peach, pear, pineapple)	Positive	Positive	Reverse
3. Legumes	Lentils, chickpeas, beans, peas	Positive	Positive	Reverse
4. Whole grains	Whole-grain bread	Positive	Positive	Reverse
5. Refined grains	White bread, rolls, white rice, white pasta	Positive	Reverse	Positive
7. Potatoes	Boiled and roasted potatoes	Positive	Positive	Reverse
6. Fries and chips	French fries, potato chips	Positive	Reverse	Positive
8. Nuts	Pine nuts, almonds, peanuts, hazelnuts, and other nuts	Positive	Positive	Reverse
9. Olive oil	Olive oil	Positive	Positive	Reverse
10. Tea and coffee	Caffeinated coffee, decaffeinated coffee, tea	Not scored	Positive	Reverse
11. Fruit juices	Orange juice, other packaged fruit juices	Not scored	Reverse	Positive
12. Sugar-sweetened beverages	Carbonated soft drinks: cola, orange, lemon	Not scored	Reverse	Positive
13. Sweets and desserts	Maria cookies, chocolate cookies, croissants, donuts, muffins, cakes, pies, churros (fried dough), chocolate, bonbons, cocoa powder, sugar	Not scored	Reverse	Positive
Animal food groups				
14. Meat/meat products	Chicken with or without skin, beef, pork, lamb, game meat (rabbit, quail, duck), liver of beef, pork or chicken, viscera, cold cuts (ham, salami, mortadella) sausages and similar, foie gras, hamburger, bacon	Reverse	Reverse	Reverse
15. Animal fats for cooking or as a spread	Butter, lard	Reverse	Reverse	Reverse
16. Eggs	Eggs	Reverse	Reverse	Reverse
17. Fish and other seafood	Fried fish, boiled or grilled fish (hake, sole, sardines, tuna), salted fish (cod, anchovies), canned fish (tuna, sardines, herring), clams, mussels, oysters, squid, octopus, shellfish (prawns, lobster and similar)	Reverse	Reverse	Reverse
18. Dairy products	Whole milk, skim or low-fat milk, condensed milk, yoghurt, cottage cheese, curd, white or fresh cheese, creamy cheese or cheese in portions, cured or semi-cured cheese (Manchego), custard, flan, pudding, ice cream	Reverse	Reverse	Reverse

Abbreviations: gPVG, general pro-vegetarian food pattern; hPVG, healthful pro-vegetarian food pattern; uPVG, unhealthful pro-vegetarian food pattern. ^1^ Positive indicates that higher consumption of this food group received higher scores. The reverse indicates that higher consumption of this food group received lower scores. ^2^ In the hPVG food pattern, whole grains, fruits, vegetables, nuts, legumes, potatoes (boiled and roasted), tea, and coffee were considered “healthy plant foods.” Refined grains, French fries and chips, fruit juices, sugar-sweetened beverages, and sweets and desserts were considered “unhealthy plant foods.” The gPVG food pattern did not differentiate plant foods as healthy or unhealthy. ^3^ In the gPVG food pattern, consumption of whole grains and refined grains were considered as the “grains” group, and potatoes and fries and chips were considered as the “potatoes” group.

**Table 2 nutrients-14-05288-t002:** Sociodemographic characteristics, lifestyle, and pro-vegetarian food patterns among controls and cancer cases (oesophagus, stomach, and pancreas) of the PANESOES study (*n* = 1233).

*N*º of Participants (%)	Controls 455 (36.9)	Cases 778 (63.1)		
		Oesophagus 199 (25.6)	Stomach 414 (53.2)	Pancreas 165 (21.2)
Age (years)	63.0 (10.7) ^1^	60.5 (9.8)	64.8 (11.4)	65.2 (11.6)
Sex, *n* (%)				
Male	285 (62.6)	184 (92.5)	271 (65.5)	100 (60.6)
Female	170 (37.4)	15 (7.5)	143 (34.5)	65 (39.4)
Province, *n* (%)				
Valencia	316 (69.5)	154 (77.4)	281 (67.9)	105 (63.6)
Alicante	139 (30.5)	45 (22.6)	133 (32.1)	60 (36.4)
Educational level, *n* (%)				
<Primary	246 (54.1)	112 (56.3)	250 (60.3)	90 (54.5)
Primary	172 (37.8)	66 (33.2)	129 (31.2)	56 (33.9)
>Primary	37 (8.1)	21 (10.6)	35 (8.5)	19 (11.5)
Alcohol drinking, *n* (%)				
Never	183 (40.2)	17 (8.5)	146 (35.3)	56 (33.9)
1–24 g/day	162 (35.6)	36 (18.1)	132 (31.9)	58 (35.2)
25–49 g/day	50 (11.0)	24 (12.1)	64 (15.5)	14 (8.5)
50–99 g/day	41 (9.0)	53 (26.6)	47 (11.4)	25 (15.2)
>99 g/day	19 (4.2)	69 (34.7)	25 (6.0)	12 (7.3)
Energy intake (Kcal/day)	1800.8 (620.5)	2274.3 (821.2)	1996.8 (662.5)	1934.4 (741.6)
Tobacco smoking, *n* (%)				
Never	218 (47.9)	23 (11.6)	174 (42.0)	72 (43.3)
Former	117 (25.7)	54 (27.1)	92 (22.1)	34 (20.7)
≤24 c/day	87 (19.1)	58 (29.1)	106 (25.7)	37 (22.6)
>24 c/day	33 (7.3)	64 (32.2)	42 (10.2)	22 (13.4)
gPVG (points of score)	36.8 (5.3)	34.6 (5.6)	35.7 (5.1)	35.8 (5.4)
hPVG (points of score)	53.9 (7.1)	55.3 (6.5)	53.4 (6.6)	54.2 (6.5)
uPVG (points of score)	53.6 (5.6)	54.5 (6.4)	54.4 (6.0)	53.0 (6.3)

Abbreviations: SD, Standard Deviation; c/day, cigarettes per day; gPVG, general pro-vegetarian food pattern; hPVG, healthful pro-vegetarian food pattern; uPVG, unhealthful pro-vegetarian food pattern. ^1^ Mean (SD) (all such values).

**Table 3 nutrients-14-05288-t003:** Association between adherence to general PVG food pattern in quintiles and continuous (per 1 unit) and oesophageal, stomach, and pancreas cancer in participants of the PANESOES study (*n* = 1233).

gPVG Food Pattern Quintiles							
	Very Low: <32	Low: 32–35	Moderate: 36–37	High: 38–41	Very High: >41	Per 1 Unit Increment in Adherence	*p*-Trend ^3^
		RRR (95% CI)	RRR (95% CI)	RRR (95% CI)	RRR (95% CI)	RRR (95% CI)	
Oesophagus, *n*	58	56	27	40	18	199	
Model 1 ^1^	Ref.	0.69 (0.42, 1.12)	0.55 (0.31, 1.00)	0.41 (0.24, 0.69)	0.31 (0.16, 0.60)	0.93 (0.89, 0.96)	0.001
Model 2 ^2^	Ref.	0.72 (0.56, 0.93)	0.60 (0.51, 0.71)	0.47 (0.38, 0.59)	0.37 (0.32, 0.42)	0.94 (0.91, 0.97)	0.01
Stomach, *n*	86	115	67	98	48	414	
Model 1 ^1^	Ref.	0.82 (0.55, 1.25)	0.85 (0.53, 1.37)	0.55 (0.36, 0.84)	0.38 (0.23, 0.64)	0.94 (0.92, 0.97)	0.001
Model 2 ^2^	Ref.	0.78 (0.63, 0.97)	0.88 (0.69, 1.13)	0.53 (0.42, 0.65)	0.34 (0.27, 0.43)	0.94 (0.92, 0.96)	0.001
Pancreas, *n*	41	42	19	37	26	165	
Model 1 ^1^	Ref.	0.61 (0.36, 1.04)	0.49 (0.26, 0.94)	0.41 (0.24, 0.71)	0.39 (0.21, 0.74)	0.94 (0.91, 0.98)	0.001
Model 2 ^2^	Ref.	0.66 (0.51, 0.84)	0.53 (0.45, 0.62)	0.46 (0.35, 0.60)	0.43 (0.35, 0.52)	0.95 (0.92, 0.98)	0.01

Abbreviations: RRR: Relative Risk Ratio; CI: Confidence Intervals; ^1^ Adjusted for age (years), sex (male; female), and province (Alicante; Valencia); ^2^ Multivariable model of the multinomial logistic regression adjusted by age (years), sex (male; female), province (Alicante; Valencia), education level (<Primary; Primary; ≥Secondary), tobacco consumption (Never; Former smoker; ≤24 c/day; ≥25 c/day), alcohol intake (never; 1–24 g/d; 25–49 g/d; 50–99 g/d; ≥100 g/d), and energy intake; ^3^ *p*-value from trend test.

**Table 4 nutrients-14-05288-t004:** Association between adherence to healthful PVG food pattern in quintiles and continuous (per 1 unit) and oesophageal, stomach, and pancreas cancer in participants of the PANESOES study (*n* = 1233).

hPVG Food Pattern Quintiles							
	Very Low <48	Low: 49–52	Moderate: 53–56	High: 57–60	Very High: >60	Per 1 Unit Increment in Adherence	*p*-Trend ^3^
		RRR (95% CI)	RRR (95% CI)	RRR (95% CI)	RRR (95% CI)	RRR (95% CI)	
Oesophagus, *n*	29	36	50	42	42	199	
Model 1 ^1^	Ref.	1.55 (0.87, 2.76)	2.57 (1.47, 4.51)	1.97 (1.10, 3.52)	2.18 (1.19, 4.03)	1.04 (1.01, 1.07)	0.01
Model 2 ^2^	Ref.	1.09 (0.87, 1.37)	1.26 (1.02, 1.56)	0.87 (0.70, 1.08)	0.72 (0.58, 0.90)	0.98 (0.95, 1.00)	0.10
Stomach, *n*	92	89	102	73	58	414	
Model 1 ^1^	Ref.	1.15 (0.77, 1.72)	1.56 (1.03, 2.36)	0.96 (0.62, 1.49)	0.70 (0.44, 1.14)	0.98 (0.96, 1.00)	0.10
Model 2 ^2^	Ref.	0.99 (0.79, 1.24)	1.18 (0.97, 1.45)	0.66 (0.54, 0.82)	0.42 (0.34, 0.52)	0.95 (0.94, 0.97)	0.01
Pancreas, *n*	32	35	37	32	29	165	
Model 1 ^1^	Ref.	1.31 (0.75, 2.28)	1.63 (0.92, 2.87)	1.22 (0.67, 2.20)	1.00 (0.53, 1.90)	1.00 (0.97, 1.03)	0.10
Model 2 ^2^	Ref.	1.17 (0.91, 1.52)	1.38 (1.10, 1.73)	0.96 (0.77, 1.21)	0.74 (0.59, 0.92)	0.98 (0.96, 1.00)	0.10

Abbreviations: RRR: Relative Risk Ratio; CI: Confidence Intervals; ^1^ Adjusted for age (years), sex (male; female), and province (Alicante; Valencia); ^2^ Multivariable model of the multinomial logistic regression adjusted by age (years), sex (male; female), province (Alicante; Valencia), education level (<Primary; Primary; ≥Secondary), tobacco consumption (Never; Former smoker; ≤24 c/day; ≥25 c/day), alcohol intake (never; 1–24 g/d; 25–49 g/d; 50–99 g/d; ≥100 g/d), and energy intake. ^3^ *p*-value from trend test.

**Table 5 nutrients-14-05288-t005:** Association between adherence to unhealthful PVG food pattern in quintiles and continuous (per 1 unit) and oesophageal, stomach, and pancreas cancer in participants of the PANESOES study (*n* = 1233).

uPVG Food Pattern Quintiles							
	Very low <50	Low: 50–52	Moderate: 53–56	High: 57–59	Very High: >59	Per 1 Unit Increment in Adherence	*p*-Trend ^3^
		RRR (95% CI)	RRR (95% CI)	RRR (95% CI)	RRR (95% CI)	RRR (95% CI)	
Oesophagus, *n*	42	33	39	40	45	199	
Model 1 ^1^	Ref.	0.88 (0.51, 1.53)	0.75 (0.45, 1.27)	1.14 (0.66, 1.95)	1.56 (0.91, 2.66)	1.02 (0.99, 1.05)	0.10
Model 2 ^2^	Ref.	0.92 (0.74, 1.15)	0.67 (0.54, 0.84)	0.97 (0.73, 1.28)	1.26 (1.00, 1.60)	1.01 (0.99, 1.03)	0.10
Stomach, *n*	86	62	112	74	80	414	
Model 1 ^1^	Ref.	0.92 (0.60, 1.42)	1.25 (0.85, 1.84)	1.29 (0.84, 1.98)	1.53 (1.00, 2.36)	1.02 (1.00, 1.05)	0.05
Model 2 ^2^	Ref.	0.99 (0.79, 1.24)	1.37 (1.12, 1.68)	1.42 (1.13, 1.78)	1.76 (1.42, 2.18)	1.03 (1.02, 1.05)	0.01
Pancreas, *n*	51	33	36	16	29	165	
Model 1 ^1^	Ref.	0.84 (0.50, 1.41)	0.69 (0.42, 1.15)	0.48 (0.26, 0.91)	0.94 (0.54, 1.64)	0.98 (0.96, 1.01)	0.10
Model 2 ^2^	Ref.	0.86 (0.66, 1.11)	0.66 (0.52, 0.84)	0.48 (0.40, 0.59)	0.91 (0.72, 1.14)	0.98 (0.96, 1.00)	0.10

Abbreviations: RRR: Relative Risk Ratio; CI: Confidence Intervals; ^1^ Adjusted for age (years), sex (male; female), and province (Alicante; Valencia); ^2^ Multivariable model of the multinomial logistic regression adjusted by age (years), sex (male; female), province (Alicante; Valencia), education level (<Primary; Primary; ≥Secondary), tobacco consumption (Never; Former smoker; ≤24 c/day; ≥25 c/day), alcohol intake (never; 1–24 g/d; 25–49 g/d; 50–99 g/d; ≥100 g/d), and energy intake. ^3^ *p*-value from trend test.

## Data Availability

The data presented in this study are available on request from the corresponding author. The data are not publicly available due to confidentiality and ethical reasons.

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
