# Peer review of "Adherence to Pro-Vegetarian Food Patterns and Risk of Oesophagus, Stomach, and Pancreas Cancers: A Multi Case–Control Study (The PANESOES Study)"

_nutrients, 2022, doi:10.3390/nu14245288_

Round 1

Reviewer 1 Report

Thank you for the opportunity to review your article.

This study is a multi case–control study that assesses the  association between three defined pro–vegetarian  food patterns and the cancers of oesophagus, stomach and pancreas.

The link between aflatoxin exposure and oesophageal cancer was not mentioned in the introduction; however aflatoxins contaminate a large proportion of foods, including maize, cereals, peanuts and tree nuts. These mycotoxins contaminates foods that can be included in a healthy pro-vegetarian diet.

The discussion section is too general.

In this section you mention the benefits of fibre, but how does this relate to the 3 cancers? Food ingredients and their mechanisms linked to the chosen cancers should be analysed!

            The main problem with the manuscript is that it fails to propose dietary recommendations to exploit the effect of plant-based nutrients on the discussed  cancers.

Author Response

Thank you very much.

Reviewer 2 Report

 The manuscript “Adherence to pro–vegetarian food patterns and risk of esophagus, stomach and pancreas cancers: a multi case–control study (The PANESOES study)” is very interesting, well-conceived and well written.

Corrections I would suggest:

1.      Line 60 – “difficulty in studying” instead of “limitation of”

2.      Lines 68-74 – You should explain better the concept of PVG and 3 different groups. References 16 and 17 should be: Satija et al, Journal of the American College of cardiology, 2017 and Kim et al, PlosMED, 2020.

3.      Important: Lines 162-167: Did you collect some data on diseases, BMI or other indicators of nutritional status? It would be important to include these data in your analyzes also.

4.      Discussion: the first sentence should be written better in English. The whole discussion can be improved. Lines from 247 are general observations and should be placed at the beginning, after the summary of results. All limitations and strengths should be grouped together at the end.

Author Response

Thank you very much.
